# State-of-the-Art Native Mass Spectrometry and Ion Mobility Methods to Monitor Homogeneous Site-Specific Antibody-Drug Conjugates Synthesis

**DOI:** 10.3390/ph14060498

**Published:** 2021-05-24

**Authors:** Evolène Deslignière, Anthony Ehkirch, Bastiaan L. Duivelshof, Hanna Toftevall, Jonathan Sjögren, Davy Guillarme, Valentina D’Atri, Alain Beck, Oscar Hernandez-Alba, Sarah Cianférani

**Affiliations:** 1Laboratoire de Spectrométrie de Masse BioOrganique, IPHC UMR 7178, Université de Strasbourg, CNRS, 67087 Strasbourg, France; evolene.desligniere@etu.unistra.fr (E.D.); anthony.ehkirch@novartis.com (A.E.); ahernandez@unistra.fr (O.H.-A.); 2Infrastructure Nationale de Protéomique ProFI—FR2048, 67087 Strasbourg, France; 3School of Pharmaceutical Sciences, University of Geneva, CMU—Rue Michel-Servet 1, 1211 Geneva, Switzerland; Bastiaan.Duivelshof@unige.ch (B.L.D.); Davy.Guillarme@unige.ch (D.G.); Valentina.Datri@unige.ch (V.D.); 4Institute of Pharmaceutical Sciences of Western Switzerland, University of Geneva, CMU—Rue Michel-Servet 1, 1211 Geneva, Switzerland; 5Genovis AB, SE-220 07 Lund, Sweden; hanna.toftevall@genovis.com (H.T.); jonathan.sjogren@genovis.com (J.S.); 6IRPF—Centre d’Immunologie Pierre-Fabre (CIPF), 74160 Saint-Julien-en-Genevois, France; alain.beck@pierre-fabre.com

**Keywords:** native mass spectrometry, size-exclusion chromatography (SEC), ion mobility-mass spectrometry (IM-MS), collision-induced unfolding (CIU), antibody-drug conjugate (ADC), site-specific conjugation

## Abstract

Antibody-drug conjugates (ADCs) are biotherapeutics consisting of a tumor-targeting monoclonal antibody (mAb) linked covalently to a cytotoxic drug. Early generation ADCs were predominantly obtained through non-selective conjugation methods based on lysine and cysteine residues, resulting in heterogeneous populations with varying drug-to-antibody ratios (DAR). Site-specific conjugation is one of the current challenges in ADC development, allowing for controlled conjugation and production of homogeneous ADCs. We report here the characterization of a site-specific DAR2 ADC generated with the GlyCLICK three-step process, which involves glycan-based enzymatic remodeling and click chemistry, using state-of-the-art native mass spectrometry (nMS) methods. The conjugation process was monitored with size exclusion chromatography coupled to nMS (SEC-nMS), which offered a straightforward identification and quantification of all reaction products, providing a direct snapshot of the ADC homogeneity. Benefits of SEC-nMS were further demonstrated for forced degradation studies, for which fragments generated upon thermal stress were clearly identified, with no deconjugation of the drug linker observed for the T-GlyGLICK-DM1 ADC. Lastly, innovative ion mobility-based collision-induced unfolding (CIU) approaches were used to assess the gas-phase behavior of compounds along the conjugation process, highlighting an increased resistance of the mAb against gas-phase unfolding upon drug conjugation. Altogether, these state-of-the-art nMS methods represent innovative approaches to investigate drug loading and distribution of last generation ADCs, their evolution during the bioconjugation process and their impact on gas-phase stabilities. We envision nMS and CIU methods to improve the conformational characterization of next generation-empowered mAb-derived products such as engineered nanobodies, bispecific ADCs or immunocytokines.

## 1. Introduction

In the last decade, antibody-drug conjugates (ADC) have evolved into promising and efficient therapeutic agents for targeted chemotherapy, with 9 ADCs currently approved by the Food and Drug Administration (FDA), and more than 80 in clinical studies [1]. ADCs are generated through the conjugation of monoclonal antibodies (mAbs) which specifically target the tumor cell, with highly potent cytotoxic drug payloads. Both elements are covalently bound via a cleavable or non-cleavable chemical linker. First-generation ADCs suffered from insufficient potency of the payload or toxicity due to the instability of the ADC, leading to premature drug release [2]. Extensive development efforts led to second-generation ADCs, with more potent payloads, improved linker stability and lower levels of unconjugated mAbs [3]. Bestselling second-generation ADCs include brentuximab vedotin (BV, Adcetris^®^ from Seattle Genetics) and trastuzumab emtansine (T-DM1, Kadcyla^®^ from Roche) [4]. However, challenges remain for these ADCs, most notably related to product heterogeneity. Drug conjugation typically occurs through primary amines of lysine side-chains (T-DM1) or cysteine thiol groups after reduction of the interchain disulfide bonds (BV). The conjugation process results in a heterogeneous mixture of species ranging from 0 to 8 payload molecules per antibody, with average drug-to-antibody ratios (avDAR) of 3–4. Drawbacks of the second-generation ADCs include competition with unconjugated mAbs, but also fast clearance and possible aggregation of high DAR species [5,6].

Building on lessons learned from past-generations products, several strategies to produce more homogeneous site-specific ADCs with improved pharmacokinetics have been developed [3,7,8], including the addition of engineered cysteine residues at specific sites [7,8,9,10,11], the use of microbial transglutaminases to attach amine-containing payloads to glutamine residues in the antibody backbone, thus connecting the drug to the antibody via a stable amide linkage [12,13,14], and the introduction of unnatural amino acids to provide a chemical handle on their conjugation [15,16]. As an alternative, recent development of new heterobifunctional reagents for maleimide conjugations were also described to produce homogeneous site-specific ADCs [17,18]. Among the different approaches that can be used to generate homogeneous ADCs, glycan-mediated conjugation based on the Asn297 residue appears as an appealing alternative [19]. The glycan moiety contained in the Fc region of mAbs can be modified through different engineering strategies to accommodate cargo molecules and produce homogeneous site-specific ADCs [19,20]. We used this technology developed by van Geel et al. to generate a custom-made DAR2 ADC with two drugs per antibody [19]. The conjugation uses a three-step procedure, consisting of deglycosylation, azide activation and click reaction (Figure 1). Specifically, the deglycosylation step allows the glycans to be trimmed after the innermost N-acetylglucosamine (GlcNAc) glycan moiety. Then, the addition of an N-azidoacetylgalactosamine (GalNAz) is performed through the azide activation step. As result, the azido-modified glycans become site-specifically reactive for copper-free click reaction with any alkyne containing payload of choice. By applying this strategy, the drug stoichiometry is controlled in a site-specific manner and localized on the Fc region of the mAb, while the antibody-binding region (Fab) is preserved, and thus, minimal influence on the immunoreactivity is expected.

The development and optimization of ADCs involve in-depth analytical and bioanalytical characterization along the production process, to monitor several critical quality attributes, such as the drug load distribution (DLD), the amount of unconjugated antibody (D0), the avDAR ratio and the presence of size variants [21,22]. State-of-the-art approaches for ADC analysis comprise chromatographic, electrophoretic and mass spectrometric techniques [21,23]. Among them, native mass spectrometry (nMS), which retains noncovalent assemblies, has now entered into R&D laboratories. Valliere-Douglass et al. first highlighted the benefits of nMS for intact mass measurement and relative distribution of drug-loaded species in the case of cysteinyl-linked ADCs [24]. Chen et al. described successful use of nanoESI instead of conventional ESI for cysteine-linked ADCs after proteolytic drug removal [25]. Online coupling of size exclusion chromatography (SEC) to nMS was then implemented by different groups for the analysis of mAbs and ADCs [26,27,28,29,30], paving the way for routine integration of nMS in high throughput analytical workflows of biopharmaceutical companies. An additional level of separation can be achieved through ion mobility spectrometry coupled to nMS (nIM-MS), which provides conformational characterization in the gas phase. nIM-MS was employed for the direct determination of distribution profiles and avDAR values of second-generation ADCs, BV and T-DM1 [31,32]. Although the DAR calculation based on nIM-MS results is not as straightforward as from nMS data, the overall drift time of ADC species obtained from nIM-MS analysis allows DAR comparison in a rapid manner. Drug binding can also be assessed by measuring collision cross sections (CCS), which correspond to the momentum transfer between ion and gas particles, and represent the effective area of ions interacting with the buffer gas [30,31,32]. However, nIM-MS sometimes fails to separate co-drifting species with closely related conformations due to its low resolution, as exemplified by mAb-biotin conjugates, which exhibit only very minor CCS differences (<2%) compared to unconjugated mAbs [33]. IM-based collision-induced unfolding (CIU) approaches have proved to be efficient to circumvent poor linear travelling-wave IM (TWIMS) resolution, offering further insight into gas-phase behavior upon ion activation in the instrument trap cell [34]. Destabilization of biotinylated model ADCs was detected with CIU even for low amounts of conjugated biotin, highlighting the potential of CIU to tackle small conformational changes between the ADC and its parent mAb [33]. Another study performed on a site-specific DAR4 ADC evidenced increased resistance to gas-phase unfolding of the ADC compared to its unconjugated counterpart mAb [30]. Few papers have reported the characterization of ADCs using CIU, most likely because of the heterogeneity of early-generation ADCs, yet this technique can provide valuable information to evaluate the gas-phase stabilization or destabilization along the conjugation process. 

We highlight in this study the potential of last generation cutting edge nMS and IM methodologies for the characterization a customized DAR2 trastuzumab–GlyCLICK–DM1 (T-GlyCLICK-DM1) generated through glycan-based enzymatic remodeling and click chemistry. SEC-nMS allows thorough identification and quantification of the different species involved either during the synthesis or in the context of forced degradation studies. Innovative IM-based CIU approaches were used to monitor the modifications in the unfolding pattern of the different conjugational intermediates isolated during T-GlyCLICK-DM1 formation. The combination of SEC-nMS and gas-phase CIU experiments provided better characterization of ADCs, affording new techniques to monitor the binding, gas-phase and conformational stabilities of the different intermediates during the conjugation process (Figure 2). IM-based CIU experiments presented in this work allow to broaden the scope of analytical information available for ADCs physicochemical characterization, from the basic assessment of the number of payloads and the drug-load distribution (SEC-nMS) to gas-phase conformational behavior. We propose here SEC-nMS and IM-based CIU methods as innovative analytical techniques, complementary to more classical biophysical techniques already implemented in most R&D laboratories, to improve the conformational characterization of next-generation empowered ADCs.

## 2. Results and Discussion

### 2.1. Online SEC-nMS to Monitor the Conjugation Process

We first investigated the initial (T0), intermediate (T1 and T2) and end (T-GlyCLICK-DM1) products using SEC-nMS, a methodology particularly well-suited not only for fast desalting of mAbs products, but also for size variant identification and quantification [27,35] (Figure 3, Table 1).

For the initial T0 compound, monomeric trastuzumab (>99.5% based on the SEC-UV chromatogram) was detected as the main compound with its glycoforms by SEC-nMS, along with the presence of very low amounts of high-molecular weight species (HMWS, peak I, Figure 3A), in agreement with previously published trastuzumab SEC-nMS analyses (Figure 3) [27]. The first step led to the formation of a main product T1 corresponding to deglycosylated trastuzumab, bearing hallmarks of the deglycosylation process through core fucose and GlcNac residues (+349 Da on each HC). Minor species corresponding to T1, with one of its heavy chain (HC) having only one GlcNac moiety attached (+203 Da), and to glycation of the T1 intermediate (+162 Da) were also detected (Figure 3B). The azide activation leads to the conversion of the triplet peak into another triplet (T2), with a mass increase of +244 Da on each HC (Figure 3B). Several additional low-molecular weight species (LMWS) were detected by SEC-UV, suggesting that the azide activation step slightly affects the stability of the mAb and forms higher amounts of LMWS, including Fc-Fab (peak III) or LC, Fd and Fab fragments (peak IV) (Figure 3A, Table 1). Finally, drug conjugation was monitored in the last step, ending up with a homogeneous peak with a mass of 148,957 ± 1 Da, corresponding to the binding of two DM1 molecules (one on each HC). T-GlyCLICK-DM1 exhibits a single avDAR2 population in agreement with the site-specific glycan-based conjugation, resulting in a straightforward SEC-nMS spectrum contrary to the highly heterogeneous and complex T-DM1 spectrum with species from D0 to D8 (Appendix A). Of note, lower amounts of LMWS were obtained for the final T-GlyCLICK-DM1 product compared to azide-activated T2, with LC, Fd or Fab fragments that were not observed on the chromatogram, but still detected by nMS which has a higher sensitivity than SEC-UV. 

Altogether, the mass accuracy of nMS combined with SEC separation allowed to unambiguously identify and quantify all products, highlighting the versatility of SEC-nMS for ADC analysis.

### 2.2. Forced Degradation Studies

To evaluate the stability of the T-GlyCLICK-DM1 product, we performed forced degradation studies at high-temperature (50 °C) for 15 days followed by SEC-nMS analysis [36].

Forced degradation studies of the final T-GlyCLICK-DM1 product revealed four main peaks on the SEC chromatogram (Figure 4A). Two main species were observed on the MS spectrum of peak II. The intact T-GlyCLICK-DM1 degraded upon thermal stress, resulting in two species with masses of 148,415 ± 10 Da (−545 Da compared to the intact product) and 147,882 ± 9 Da (−1078 Da) (Figure 4B). As no mass shifts were observed for T0, T1 and T2 after thermal stress, these two degradation products most likely correspond to the loss of maytansinol after ester hydrolysis within the DM1 drug (−548 Da) [37]. Similarly, losses of −560 and −1111 Da were detected on Fc-Fab fragments (peak III, Figure 4B). No deconjugation was observed on T-GlyCLICK-DM1, as minor species still correspond to DAR2.

Previous thermal stress studies performed on mAbs have evidenced the formation of LMWS, which result mainly from fragmentation in the hinge region, and formation of HMW aggregates [36,38]. While the aggregation and hinge-fragmentation of therapeutic mAbs have been extensively studied [39,40,41], only few papers have dealt with stressed ADCs, focusing mainly on their aggregation, but lacking a detailed characterization of LMWS [42,43]. Wakankar et al. showed, using SEC analysis, that T-DM1 was more prone to aggregation than unconjugated trastuzumab, which was further emphasized after storage at 40 °C for 70 days [44]. Temperature-induced aggregation as a function of increasing DAR was also examined for a cysteine-linked ADC, highlighting that high DAR species were far more likely to form aggregates under stressed conditions [45]. 

For the GlyCLICK conjugation process, higher amounts of HMWS and LMWS are generated for the initial, intermediate and final reaction products upon thermal stress. Additional LMWS (peak IV) corresponding to LC, Fab and Fd fragments that were not observed on the SEC-UV chromatograms of non-stressed samples (expect for T2, Figure 3) were detected (Figure 4A and Appendix A). In particular, for T-GlyCLICK-DM1 (Figure 4), an increased amount of HMWS corresponding to dimers was detected for the thermally-stressed sample compared to the non-stressed one (peak I, 4.6 vs. 0.2%, respectively). Regarding LMWS, the fraction of Fc-Fab species (peak III) significantly increased upon thermal stress (+9.1%), and a substantial amount of Fab, LC and Fd fragments could be observed (peak IV, 4.6%).

Of note, different species were identified as Fab fragments, with a ladder of cleavage sites on the HC upper hinge sequence C^223^/D/K/T/H/T/C^229^, as already reported for IgG1 mAbs [40,41]. These Fab fragments have been described as a result of direct hydrolysis of peptide bonds, or radical transfer between the aforementioned residues [46,47,48]. Other LMW species detected within peak IV include a Asp^1^-Glu^213^ LC fragment and a Glu^1^-Ser^222^ Fd fragment, generated after cleavage of the HC-LC disulfide bond. The scission of the Cys^223^-Cys^214^ bond can occur either via β-elimination [48] or via a radical reaction mechanism [47]. The presence of sulfurized cysteines following the disruption of the Cys^223^-Cys^214^ bond was also previously demonstrated (+32 Da, Figure 4B) [47]. These different cleavage products were observed for all products of the GlyCLICK reaction (Appendix A). The amount of LMW cleavage products was significantly enhanced under thermal stress; however, some species were also detected for non-stressed T2 (Table 1). Interestingly, no deconjugation of the drug linker was detected, as DAR2 species are mostly detected on intact T-GlyCLICK-DM1.

Overall, SEC-nMS allows to monitor the formation of HMW aggregates and LMW hinge-related species for all our reaction compounds subjected to thermal stress conditions. Upon thermal stress, the final T-GlyCLICK-DM1 produces higher amounts of HMWS (+4.4% compared to the non-stressed sample) and LMWS (+13.6%) than the initial product T0 (+1.2% for HMWS and +9.3% for LMWS). T-DM1 exhibits higher resistance to thermally induced fragmentation (+2.4% of LMWS) compared to T-GlyCLICK-DM1 and T0, but is more prone to aggregation (+15.0%), in agreement with conclusions published on unconjugated trastuzumab vs. T-DM1 using SEC-UV analysis [44,49] (Appendix A). However, SEC as a standalone technique does not provide sufficient information on the nature of the degradation products. Our results show a clear benefit of the SEC-nMS coupling, which offers both quantification and identification of fragments in a straightforward way, within a single run.

### 2.3. nIM-MS to Monitor the Conformational Landscape during the Conjugation Process

We next used IM-based methodologies to investigate conformational changes upon the drug conjugation process.

We first performed a ^TW^CCS_N2_ calculation on both intact and IdeS-digested conjugation compounds (Appendix A). Based on mass-derived CCS predictions of intact products, only very slight differences (<1.3%) that fall within the mass error of the IM measurement (2%) might be observed between all species under investigation. Indeed, at the intact level, differences in ^TW^CCS_N2_ were between 0.3 and 1.2% for the 23+ charge state. Middle-up level measurements provide slightly higher ^TW^CCS_N2_ variations for the Fc fragment (between 0.6 and 4.2%), which correspond to mass-related differences. ^TW^CCS_N2_ values obtained for the F(ab’)_2_ subdomain was similar for all products, as conjugation sites are located on Fc fragments. Altogether, these results suggest that the chemical conjugation process does not drastically affect the overall global conformation of the mAb. However, drawing clear-cut conclusions solely from nIM-MS measurement for mAbs with very close conformations remains challenging at both intact and middle-up levels due to the low resolution of linear TWIMS [50]. The preliminary CCS measurements are the rationale for performing further CIU experiments, as an alternative to tackle small conformational variations that would result in differences in CIU patterns.

CIU experiments were then performed on two different charge states (24+ and 23+) of the reaction products obtained along the drug conjugation process, with the aim to end up with different unfolding patterns. CIU patterns of T0 to T-GlyCLICK-DM1 are represented in Figure 5. For the 24+ charge state, the CIU fingerprint of glycosylated trastuzumab T0 reveals three unfolding transitions (four conformational states) in the 0–200 V range (Figure 5A). After the first deglycosylation step, three transitions are still detected for the T1 intermediate (Figure 5B). While the first one occurs at the same voltage for T0 and T1 (32.7 V), the second transition exhibits lower CIU50 values for T1 (57.1 V) than for glycosylated T0 (66.6 V) and the third transition happens at 177.8 V for T1, but only at 192.6 V for T0. As previously reported using CIU experiments [51,52], these results indicate that deglycosylated trastuzumab T1 is more prone to unfolding than its glycosylated counterpart, also in agreement with hydrogen-deuterium exchange (HDX) data showing increased deuterium uptake after EndoS2 deglycosylation [52]. Upon azide activation, the CIU fingerprint still looks very similar, but with slightly higher CIU50 values for the first and second transitions (37.7 and 72.7 V, respectively) (Figure 5C). The third transition (at high voltages) is not detected for T2 using automated CIU50 analysis, as the most unfolded state only starts appearing, with state 2 remaining the most intense feature until 200 V. CIU50 values suggest that the conformational states of azide-activated T2 are more resistant towards unfolding than T0 and T1, in favor of a gas-phase stabilization just before the click chemistry reaction. Finally, the conjugation of the DM1 drug on T2 confers a better gas-phase resistance to unfolding to the end product T-GlyCLICK-DM1, with two conformational transitions occurring at higher CIU50 values (42.7 and 82.4 V) than the other reaction compounds, suggesting that the click chemistry step mostly contributes to the increased resistance to unfolding of T-GlyCLICK-DM1 (Figure 5D). Similarly, CIU fingerprints of the 23+ charge state illustrate the improved stability towards unfolding of the final product compared to T0, T1 and T2 (Appendix A).

Altogether, these results highlight that drug conjugation reinforces the overall stability of the mAb towards gas-phase unfolding, as already reported for a DAR4 site-specific ADC [30].

## 3. Materials and Methods

### 3.1. Sample Preparation

T-DM1 was N-deglycosylated by incubating one unit of IgGZERO (Genovis, Lund, Sweden) per microgram of ADC for 30 min at 37 °C. For middle-up nIM-MS level experiments, IdeS digestion was performed by incubating one unit of FabRICATOR enzyme (Genovis, Lund, Sweden) per microgram of mAb or ADC for 60 min at 37 °C. 

### 3.2. Manual Buffer Exchange

Prior to nIM-MS, products T0, T1, T2 and T-GlyCLICK-DM1 were desalted against 100 mM ammonium acetate (pH 6.9), using eight cycles of centrifugal concentrator with 10 and 50 kDa cutoffs for IdeS-digested and intact mAbs, respectively (Vivaspin, Sartorius, Göttingen, Germany). Protein concentration was determined by UV absorbance using a NanoDrop spectrophotometer (Thermo Fisher Scientific, France). Each solution was diluted in 100 mM ammonium acetate at pH 6.9 to 10 µM prior to nIM-MS and CIU acquisitions.

### 3.3. Online SEC-nMS

An Acquity UPLC H-class system (Waters, Wilmslow, UK) composed of a quaternary solvent manager, a sample manager set at 10 °C, a column oven and a TUV detector operating at 280 nm and 214 nm was coupled to a Synapt G2 HDMS mass spectrometer (Waters, Wilmslow, UK) for online SEC-nMS experiments. The SEC column used was an Acquity BEH SEC 200 Å, 1.7 µm, 4.6 × 150 mm (Waters). The separation was carried out in isocratic mode with a 100 mM AcONH_4_ mobile phase at pH 6.9. The Synapt G2 was operated in positive ionization mode with a capillary voltage of 3 kV and a sample cone voltage of 180 V. The backing pressure of the Z-Spray source was set to 6 mbar. Acquisitions were performed in the 1000–10,000 *m*/*z* range. External calibration was performed using singly charged ions produced by a 2 g/L solution of cesium iodide in 2-propanol/water (50/50 *v*/*v*). SEC-nMS data interpretations were performed using MassLynx v4.1 (Waters, Manchester, UK).

### 3.4. nIM-MS and CIU Experiments

The Synapt G2 HDMS was coupled to the automated chip-based nanoESI device (TriVersa NanoMate, Advion, Ithaca, NY, USA). The cone voltage of the Synapt G2 was fixed to 80 V to avoid in-source ion activation while ensuring ion transmission. The backing pressure was 6 mbar. The argon flow rate was set to 5 mL/min. Ions were focused in the helium cell (120 mL/min), prior to IM separation. The N_2_ flow rate in the IM cell was 60 mL/min. The wave height and velocity were fixed to 40 V and 850 m/s, respectively. Drift times were converted into CCS values using avidin (for middle-up level data), concanavalin A, alcohol dehydrogenase and pyruvate kinase (for intact-level data) as external calibrants [53,54]. ATDs were extracted using MassLynx v4.1. 

CIU experiments were carried out by increasing the collision voltage in the trap cell from 0 to 200 V using steps of 5 V. CIU data were processed using the CIUSuite 2 v2.2 software [55]. ATDs were smoothed using a Savitsky-Golay algorithm with a window length of 5 and a polynomial order of 2. CIU acquisitions were performed in triplicate to generate averaged CIU fingerprints with their associated RMSD using the ‘Basic Analysis’ module of the CIUSuite 2 software. RMSDs under 15% between technical replicates account for a good reproducibility of CIU data (Appendix A). CIU50 values, which allow to quantitatively assess unfolding transitions, were determined with the ‘Stability Analysis’ module.

## 4. Conclusions

This study clearly highlights the benefits of using innovative nMS and IM methodologies for the analytical characterization of ADCproducts. In the present work, a customized homogeneous site-specific ADC generated through glycan-based enzymatic remodeling and click chemistry was used as a case study. 

First, the combination of SEC with nMS was found to be particularly well suited to monitor the ADC conjugation process. Indeed, thanks to an excellent mass accuracy and sensitivity, the characterization and quantification of the different reaction products (intermediates) obtained during the drug conjugation process were easily assessed. SEC-nMS was also found to be relevant in forced degradation studies, for the simultaneous identification and quantification of LMWS and HMWS within the same run. Indeed, upon thermal stress, several HMWS and LMWS were produced and clearly identified with SEC-nMS. With the site-specific ADC product investigated in this work, no deconjugation of the drug linker was detected. The SEC-nMS data emphasize the importance of the technique to accurately characterize the drug form and bioconjugation intermediates prior to moving on to in vivo studies. Based on its noticeable advantages, SEC-nMS is expected to soon become a standard in R&D biopharmaceutical laboratories [27].

Next, IM-based methodologies were used to investigate conformational changes upon the drug conjugation process. Even if CCS measurements are not highly informative on intact ADCs nor subunits obtained after protease treatment, results suggest showed that the chemical conjugation process does not drastically affect the overall global conformation of the mAb. However, drawing clear conclusions solely from CCS values was difficult due to the low resolution of linear TWIMS. Therefore, advanced innovative CIU experiments were performed to compare the resistance to gas-phase unfolding of the different intermediates observed during the conjugation process. Based on the unfolding patterns, it was possible to conclude that the drug conjugation improves the overall stability of the mAb against gas-phase unfolding, allowing to circumvent limitations of CCS measurements for mAb-based products. These results demonstrate that CIU approaches offer clear benefits over standard nIM-MS experiments to detect subtle conformational differences that translate into different CIU patterns. In addition, CIU data have been reported to correlate with unfolding patterns observed using differential scanning calorimetry (DSC), suggesting a solution-phase memory effect of mAbs products in the gas phase [33,50]. CIU offers significant benefits over DSC, with improved sensitivity and selectivity, and thus, appears as an appealing approach to acquire conjugation-dependent gas-phase stability shift information for biotherapeutics.

## Figures and Tables

**Figure 1 pharmaceuticals-14-00498-f001:**
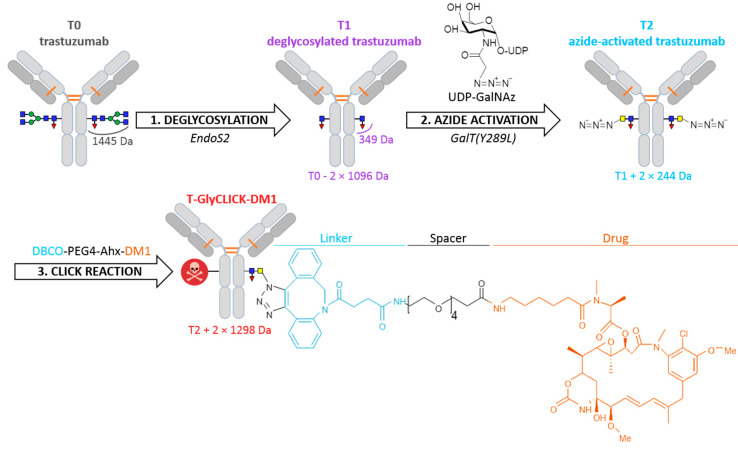
Schematic overview of the bioconjugation protocol. N-glycans remodeling of trastuzumab (T0) through deglycosylation, azide activation and click-chemistry, generating deglycosylated trastuzumab (T1), azide-activated trastuzumab (T2) and T-GlyCLICK-DM1, respectively.

**Figure 2 pharmaceuticals-14-00498-f002:**
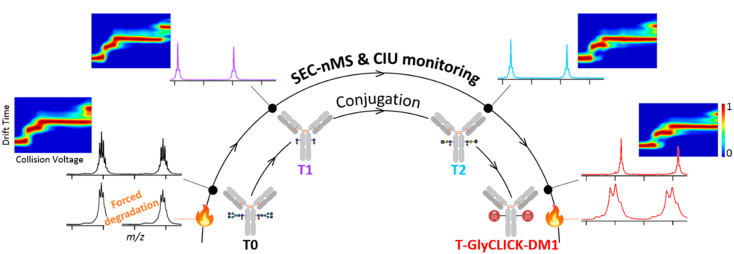
Analytical workflow used to monitor the conjugation of T-GlyCLICK-DM1.

**Figure 3 pharmaceuticals-14-00498-f003:**
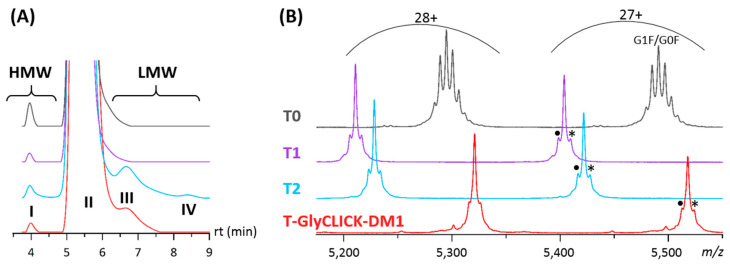
Online SEC-nMS analysis of initial (T0), intermediate (T1 and T2) and end (T-GlyCLICK-DM1) conjugation products. (**A**) Zoom on SEC-UV chromatograms at 280 nm; I = HMW dimers, II = main product, III = Fc-Fab fragments and IV = LC, Fd and Fab fragments. (**B**) Zoom on SEC-nMS spectra obtained for the different main products; ● = a-fucosylation (−146 Da), ✱ = glycation (+162 Da).

**Figure 4 pharmaceuticals-14-00498-f004:**
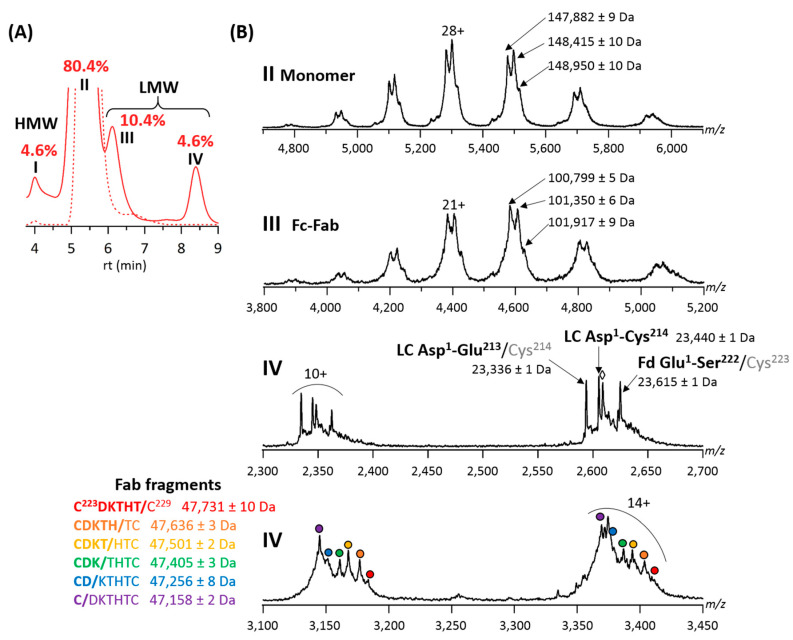
Online SEC-nMS analysis of thermally-stressed T-GlyCLICK-DM1. (**A**) Overlaid SEC chromatograms of stressed (solid line) and non-stressed (dotted line) samples. Relative amounts of HMWS and LMWS are given for the stressed sample; I = HMW dimers, II = main product, III = Fc-Fab fragments and IV = LC, Fd and Fab fragments. (**B**) SEC-nMS spectra of species generated upon thermal stress. ⬨ = sulfurized Cys^214^ (+32 Da compared to LC Asp^1^-Cys^214^).

**Figure 5 pharmaceuticals-14-00498-f005:**
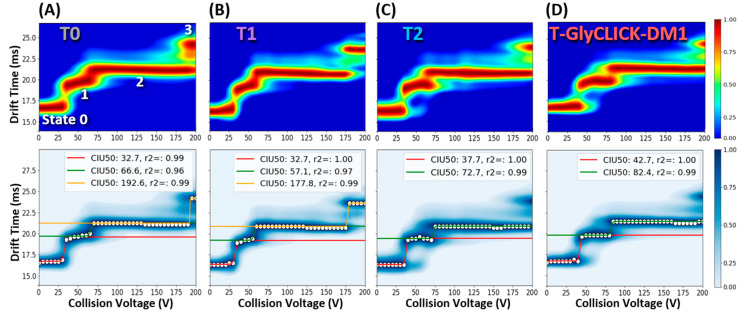
CIU experiments at the intact level for the 24+ charge state. CIU fingerprints (upper panel) and CIU50 analysis (lower panel) were acquired to compare the resistance to gas-phase unfolding of the reaction compounds (**A**) T0, (**B**) T1, (**C**) T2 and (**D**) T-GlyCLICK-DM1.

**Table 1 pharmaceuticals-14-00498-t001:** Masses of species detected with SEC-nMS. Relative quantification of the different species was assessed based on SEC-UV signals. * ND = mass not determined because of very low intensities on the nMS spectrum.

	T0	T1	T2			T-GlyCLICK-DM1		
**Main Product**	**99.6%**	**99.7%**	**96.6%**			**98.5%**		
	G0F/G0 147,930 ± 4 Da	145,875 ± 1 Da	146,372 ± 2 Da			148,957 ± 1 Da		
	(G0F)2 148,067 ± 4 Da							
	G1F/G0F 148,228 ± 2 Da							
	(G1F)2 148,387 ± 2 Da							
	G2F/G1F 148,548 ± 1 Da							
**HMW** **Dimers**	**0.4%**	**0.3%**	**0.5%**			**0.2%**		
	296,828 ± 25 Da	291,719 ± 25 Da	292,911 ± 23 Da			ND *		
**LMWS**	-	-	**2.7%**	Fc-Fab	99,319 ± 6 Da	**1.3%**	Fc-Fab	101,910 ± 8 Da
			**0.2%**	LC	23,473 ± 2 Da	**<0.1%**	LC	23,474 ± 3 Da
				Fd	23,618 ± 4 Da		Fd	23,615 ± 3 Da
				Fab	47,129 ± 9 Da		Fab	47,091 ± 8 Da

## Data Availability

The raw data presented in this study are available on request from the corresponding author.

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
