# Peer review of "State-of-the-Art Native Mass Spectrometry and Ion Mobility Methods to Monitor Homogeneous Site-Specific Antibody-Drug Conjugates Synthesis"

_pharmaceuticals, 2021, doi:10.3390/ph14060498_

Round 1

Reviewer 1 Report

  1. Novelty for development of new heterbifunctional for maleimide conjugations to end up with homogeneous site-specific ADCs. Previous studies were published in Bioconjugate Chemistry ACS in 2015. Conjugations specifically directed at Asp297 with glycan-mediated technologies is more novel with publications at recently as 2020 in high quality journals. Thus, the technology presented in the manuscript is timely, but not novel in itself. The novelty is in the application (development of a new ADC) and in the evaluation methods of the ADC in context with the glycan linker.

  1. Figure is shrunk to the extent that the font is too small to read. The GalNAz structure is very small. Figure needs to be changed so that detail better shown.

  1. For nIM-MS phase, the introduction is nice describing the evolution of MS-based interrogative systems for evaluating ADC conjugations and provide educational insight into what will be evaluated in the manuscript.

  1. T-DM1 should be a main comparator. How can the authors be certain that deconjugation will occur with T-DM1 based on their evaluation method if it was not tried?

  1. When the authors claim that SEC-nMS will soon become a standard in R&D biopharmaceutical laboratories is that in the context of accompanying CCS. Will nMS be good enough without CCS. It’s unclear what the role of CCS was to the overall value of the study, especially since CCS has poor resolution.

Author Response

Author’s response to reviewers

Title: “State of the Art Native Mass Spectrometry and Ion Mobility Methods to Monitor Homogeneous Site-Specific Antibody-Drug Conjugates Synthesis.”

Authors: Evolène Deslignière, Anthony Ehkirch, Bastiaan L. Duivelshof, Hanna Toftevall, Jonathan Sjögren, Davy Guillarme, Valentina D’Atri, Alain Beck, Oscar Hernandez-Alba and Sarah Cianférani

We thank the reviewer for his/her constructive suggestions and comments. Specific responses to each of them are given below, in red. Corresponding changes in the manuscript are highlighted in yellow in the revised version of the manuscript.

Reviewer 1

Point 1. Novelty for development of new heterobifunctional for maleimide conjugations to end up with homogeneous site-specific ADCs. Previous studies were published in Bioconjugate Chemistry ACS in 2015. Conjugations specifically directed at Asp297 with glycan-mediated technologies is more novel with publications at recently as 2020 in high quality journals. Thus, the technology presented in the manuscript is timely, but not novel in itself. The novelty is in the application (development of a new ADC) and in the evaluation methods of the ADC in context with the glycan linker.

We completely agree with the reviewer's comment. As specified in the introduction part of the paper, a general overview about the use of modified mAb glycans to produce site-specific antibodies is summarized in Qasba’s review (doi: 10.1021/acs.bioconjchem.5b00173). This work has been cited in the paper (reference 22, page 2 line 74). In addition, the specific bioconjugation used to produce the T-GlyCLICK-DM1 product was developed by Van Geel et al. in 2015 (reference 21, page 2 line 72). However, in this work we wanted to stress the use of novel, not always routinely used MS methods, like SEC-nMS and nIM-based CIU. The innovative part of this work is really to monitor by SEC-nMS and CIU methods the stability of all the bioconjugation intermediates, from trastuzumab to T-GlyCLICK-DM1, through the study of products generated upon thermal stress and the modification of the gas-phase stability. These results are of utmost importance to determine how these modifications can affect the global stability of the product, something that is not sufficiently explored in the production of new ADC formats.  The “technological” aim of the paper has been reinforced in the introduction part: “We highlight in this study the potential of last generation cutting edge nMS and IM methodologies for the characterization a customized DAR2 trastuzumab – GlyCLICK – DM1 (T-GlyCLICK-DM1) generated through glycan-based enzymatic remodeling and click chemistry. SEC-nMS allows thorough identification and quantification of the different species involved either during the synthesis or in the context of forced degradation studies. Innovative IM-based CIU approaches were used to monitor the modifications in the unfolding pattern of the different conjugational intermediates isolated during T-GlyCLICK-DM1 formation. The combination of SEC-nMS and gas-phase CIU experiments provided better characterization of ADCs, affording new techniques to monitor the binding, gas-phase and conformational stabilities of the different intermediates during the conjugation process (Figure 2). IM-based CIU experiments presented in this work allow to broaden the scope of analytical information available for ADCs physicochemical characterization, from the basic assessment of the number of payloads and the drug-load distribution (SEC-nMS) to gas-phase conformational behavior. We propose here SEC-nMS and IM-based CIU methods as innovative analytical techniques, complementary to more classical biophysical techniques already implemented in most R&D laboratories, to improve the conformational characterization of next generation empowered ADCs.” (page 4, 121-136)

Point 2. Figure is shrunk to the extent that the font is too small to read. The GalNAz structure is very small. Figure needs to be changed so that detail better shown.

Figure 1 has been modified and enlarged:

Figure 1. Schematic overview of the bioconjugation protocol. N-glycans remodeling of trastuzumab (T0) through deglycosylation, azide activation, and click-chemistry, generating deglycosylated trastuzumab (T1), azide-activated trastuzumab (T2), and T-GlyCLICK-DM1, respectively.

Point 3. For nIM-MS phase, the introduction is nice describing the evolution of MS-based interrogative systems for evaluating ADC conjugations and provide educational insight into what will be evaluated in the manuscript.

Authors would like to thank the reviewer for recognizing the quality of our work.

Point 4. T-DM1 should be a main comparator. How can the authors be certain that deconjugation will occur with T-DM1 based on their evaluation method if it was not tried?

Unfortunately, in the exceptional case of highly heterogeneous ADCs such as T-DM1, the mass spectra upon thermal degradation are very complex, providing broad mass peaks and hindering the precise identification of the different populations generated upon thermal stress:

Besides, although T-DM1 share structural similarities with the site-specific T-GlyCLICK-DM1, the conjugation process and sites are completely different. We have thus decided to remove the straight comparison between T-DM1 and T-GlyCLICK-DM1 regarding the cleavage within DM1 to avoid misleading.

The comparison of these two ADCs has been exclusively performed in the frame of HMWS, and LMWS relative intensities obtained using SEC-UV chromatograms. We have completed the “Results - Forced degradation studies” section accordingly: “Upon thermal stress, the final T-GlyCLICK-DM1 produces higher amounts of HMWS (+4.4% compared to the non-stressed sample) and LMWS (+15.0%) than the initial product T0 (+1.2% for HMWS and +9.3% for LMWS). T-DM1 exhibits higher resistance to thermally-induced fragmentation (+2.4% of LMWS) compared to T-GlyCLICK-DM1 and T0, but is more prone to aggregation (+15.0%), in agreement with conclusions published on unconjugated trastuzumab vs T-DM1 using SEC-UV analysis [46,51] (Figure S1A).” (pages 6-7, lines 215-220)

Point 5. When the authors claim that SEC-nMS will soon become a standard in R&D biopharmaceutical laboratories is that in the context of accompanying CCS. Will nMS be good enough without CCS. It’s unclear what the role of CCS was to the overall value of the study, especially since CCS has poor resolution.

Authors underpin the idea that SEC-nMS coupling can afford clear benefits to the study of therapeutic proteins in their native state for several reasons: i) reduction of sample preparation step by performing “on-line” buffer exchange, ii) increasing the throughput of nMS analysis, and iii) providing simultaneous mass analysis and relative quantitation of HMWS and LMWS. So far, the addition of IM-MS data and CCS measurements in the context of therapeutic proteins will be meaningless, due to the low resolution of commercially available IM-MS platforms.

The reason why we added the CCS measurements in the manuscript was exactly to highlight this point in a pedagogic manner, like a rationale for the need of CIU instead of CCS measurements. CIU experiments have been proposed in the literature to detect more subtle conformational changes, for instance in the case of different mAb subclasses. However, yet CIU data acquisition and interpretation require a high level of technical ability, which precludes so far its wide adoption in the R&D laboratories of biopharmaceutical companies for mAb-derived protein studies. In our previous works (doi: 10.1021/acs.analchem.0c01426) we have developed a SEC-CIU experimental set-up that enables to record the unfolding pattern of therapeutic mAbs automatically and improves the high-throughput of CIU experiments significantly (acquisition time reduced from several hours to 15 minutes). This experimental coupling facilitates the accessibility to perform CIU experiments and might pave the way to implement this technique in the R&D biopharmaceutical laboratories.      

Reviewer 2 Report

Deslignière et al describe the use of SEC-nMS and nIMS-MS to characterize the synthesis of an ADC with site-specific conjugation and DAR.  By SEC-nMS, they were able to confirm the identity of the antibody generated in each step of the synthesis.  Both low- and high-molecular weight species were observed in the experiment and could be uniquely identified using the high mass accuracy of the TOF-MS.  Through forced-degradation studies, the authors were able to track degradation products that resulted from storage at elevated temperatures, such as cleavage sites on the hinge region and fragmentation of the DM1.  nIM-MS experiments for each step of the bioconjugation showed that the DAR2 species improved the stability of the trastuzumab and intermediates while in the gas phase.  

The authors presented their study in a clear and logical way.  The Introduction was presented with relevant and fitting references.  Conclusions were made appropriately from the data that were presented in the manuscript and supported by data included in the Supplementary Material.  The Figures and Tables were clearly presented with the appropriate amount of detail to guide the reader through the experiments.  The SEC-nMS data emphasize the importance of the technique to accurately characterize the drug form and bioconjugation intermediates prior to moving on to in vivo studies.  The nIMS-MS studies convincingly demonstrate that the ADC better resists gas-phase unfolding relative to the reaction compounds.  The only question that I have is:  does the gas-phase unfolding observed in the nIMS-MS experiments have implications for solution stability of the ADC?  Please comment on how knowledge of the gas-phase stability will impact overall drug stability in practical application (such as dosing solution stability) . 

Author Response

Author’s response to reviewers

Title: “State of the Art Native Mass Spectrometry and Ion Mobility Methods to Monitor Homogeneous Site-Specific Antibody-Drug Conjugates Synthesis.”

Authors: Evolène Deslignière, Anthony Ehkirch, Bastiaan L. Duivelshof, Hanna Toftevall, Jonathan Sjögren, Davy Guillarme, Valentina D’Atri, Alain Beck, Oscar Hernandez-Alba and Sarah Cianférani

We thank the reviewer for his/her constructive suggestions and comments. Specific responses to each of them are given below, in red. Corresponding changes in the manuscript are highlighted in yellow in the revised version of the manuscript.

Reviewer 2

Deslignière et al describe the use of SEC-nMS and nIMS-MS to characterize the synthesis of an ADC with site-specific conjugation and DAR.  By SEC-nMS, they were able to confirm the identity of the antibody generated in each step of the synthesis.  Both low- and high-molecular weight species were observed in the experiment and could be uniquely identified using the high mass accuracy of the TOF-MS.  Through forced-degradation studies, the authors were able to track degradation products that resulted from storage at elevated temperatures, such as cleavage sites on the hinge region and fragmentation of the DM1.  nIM-MS experiments for each step of the bioconjugation showed that the DAR2 species improved the stability of the trastuzumab and intermediates while in the gas phase. 

The authors presented their study in a clear and logical way.  The Introduction was presented with relevant and fitting references.  Conclusions were made appropriately from the data that were presented in the manuscript and supported by data included in the Supplementary Material.  The Figures and Tables were clearly presented with the appropriate amount of detail to guide the reader through the experiments.  The SEC-nMS data emphasize the importance of the technique to accurately characterize the drug form and bioconjugation intermediates prior to moving on to in vivo studies.  The nIMS-MS studies convincingly demonstrate that the ADC better resists gas-phase unfolding relative to the reaction compounds.

Point 1. The only question that I have is:  does the gas-phase unfolding observed in the nIMS-MS experiments have implications for solution stability of the ADC?  Please comment on how knowledge of the gas-phase stability will impact overall drug stability in practical application (such as dosing solution stability)

The reviewer raises a very interesting question that has been considered as a crucial point to interpret native IM-MS results and how these results can be linked to protein structure in solution. In this case, CIU is considered a relatively budding technique where the vast majority of the data regarding the unfolding mechanism of biomolecular ions have been restricted to the gas-phase state. However, several attempts in the literature have been performed to correlate these data to the stability of proteins in solution. For instance, our group and other colleagues have been published the correlation of CIU analyses of therapeutic mAbs (doi: 10.1021/acs.analchem.0c00293) and ADCs (doi: 10.1002/pro.3560) with differential scanning calorimetry (DSC) data. In both cases, the correlation between these two techniques is not perfect, nevertheless, they provided experimental evidence indicating a strong solution-phase memory effect of the mAbs and ADCs structure in the gas-phase. Even though further analyses should be performed to provide a well-stablished correlation between gas-phase and solution stability data, CIU could bring significant benefits in comparison with solution stability techniques due to its sensitivity and selectivity to precisely obtain CIU data of specific mass-resolved species, for instance, different ADC DAR populations. This point has been added to the conclusion: “In addition, CIU data have been reported to correlate with unfolding patterns observed using differential scanning calorimetry (DSC), suggesting a solution-phase memory effect of mAb products in the gas phase [35,52]. CIU offers significant benefits over DSC, with improved sensitivity and selectivity, and thus appears as an appealing approach to acquire conjugation-dependent gas-phase stability shift information for biotherapeutics.” (page 11, lines 346-350)

Reviewer 3 Report

According to this reviewer, this work can be considered for publication in Pharmaceuticals after improving certain aspects of the manuscript.

1.Elements of scientific novelty should be presented in a detailed and convincing manner (in the last paragraph of the Introduction, and shortly in Abstract).

  1. The potential application of the performed method should be given in Abstract.

  1. I suggest that a diagram (scheme) presenting the overall procedure applied in the study should be given (in Section 2). It would help understand the details of the analytical protocol better.

  1. Application of proper quality assurance/quality control (QA/QC) procedures is vital for the

measurement results to be treated as a source of reliable analytical information. Consequently,  I suggest that a separate section devoted to QA/QC be added to the manuscript. Or at least, validation parameters with description…

  1. Masses of species with the corresponding errors (value ± SD) are presented in not good way; two significant digits of SD should be taken into consideration to present the results. Please, correct where required

Author Response

Author’s response to reviewers

Title: “State of the Art Native Mass Spectrometry and Ion Mobility Methods to Monitor Homogeneous Site-Specific Antibody-Drug Conjugates Synthesis.”

Authors: Evolène Deslignière, Anthony Ehkirch, Bastiaan L. Duivelshof, Hanna Toftevall, Jonathan Sjögren, Davy Guillarme, Valentina D’Atri, Alain Beck, Oscar Hernandez-Alba and Sarah Cianférani

We thank the reviewer for his/her constructive suggestions and comments. Specific responses to each of them are given below, in red. Corresponding changes in the manuscript are highlighted in yellow in the revised version of the manuscript.

Reviewer 3

According to this reviewer, this work can be considered for publication in Pharmaceuticals after improving certain aspects of the manuscript.

Point 1. Elements of scientific novelty should be presented in a detailed and convincing manner (in the last paragraph of the Introduction, and shortly in Abstract).

According to reviewer’s suggestions, the introduction and the abstract of the paper have been modified to highlight the novelty of our work. The last paragraph of the introduction was modified by adding the following information: “SEC-nMS allows thorough identification and quantification of the different species involved either during the synthesis or in the context of forced degradation studies. Innovative IM-based CIU approaches were used to monitor the modifications in the unfolding pattern of the different conjugational intermediates isolated during T-GlyCLICK-DM1 formation. The combination of SEC-nMS and gas-phase CIU experiments provided better characterization of ADCs, affording new techniques to monitor the binding, gas-phase and conformational stabilities of the different intermediates during the conjugation process (Figure 2). IM-based CIU experiments presented in this work allow to broaden the scope of analytical information available for ADCs physicochemical characterization, from the basic assessment of the number of payloads and the drug-load distribution (SEC-nMS) to gas-phase conformational behavior. We propose here SEC-nMS and IM-based CIU methods as innovative analytical techniques, complementary to more classical biophysical techniques already implemented in most R&D laboratories, to improve the conformational characterization of next generation empowered ADCs.” (page 4, lines 123-136)

Similarly, the last part of the abstract has been modified to include the novelty of our work: “Altogether, these state-of-the-art nMS methods represent innovative approaches to investigate drug loading and distribution of last generation ADCs, their evolution during the bioconjugation process and their impact on solution and gas-phase stabilities. We envision nMS and CIU methods to improve the conformational characterization of next generation empowered mAb-derived products such as engineered nanobodies, bispecific ADCs or immunocytokines.”

Point 2. The potential application of the performed method should be given in Abstract.

The requested information from the reviewer have been added in the abstract. Application of CIU and SEC-nMS techniques to other therapeutic formats have been proposed: “Altogether, these state-of-the-art nMS methods represent innovative approaches to investigate drug loading and distribution of last generation ADCs, their evolution during the bioconjugation process and their impact on solution and gas-phase stabilities. We envision nMS and CIU methods to improve the conformational characterization of next generation empowered mAb-derived products such as engineered nanobodies, bispecific ADCs or immunocytokines.”

Point 3. I suggest that a diagram (scheme) presenting the overall procedure applied in the study should be given (in Section 2). It would help understand the details of the analytical protocol better.

A figure presenting the overall workflow has been added at the end of the introduction:

Figure 2. Analytical workflow used to monitor the conjugation of T-GlyCLICK-DM1.

Point 4. Application of proper quality assurance/quality control (QA/QC) procedures is vital for the measurement results to be treated as a source of reliable analytical information. Consequently, I suggest that a separate section devoted to QA/QC be added to the manuscript. Or at least, validation parameters with description…

Different protocols based on electrophoretic, chromatographic, and mass spectrometric techniques, in combination or individually, are well-stablished in GMP environments to assess different critical quality attributes (CQAs) associated to ADCs such as the D0, DLD, the average DAR and the size variants. Further information regarding the characterization of the CQAs of ADCs is out of the scope of the current work. The main idea is to show the possibilities of state-of-the-art mass spectrometry and ion mobility techniques to provide an extensive characterization of the bioconjugation process of a site-specific ADC that could eventually be applied to different therapeutic protein formats. The authors have included two recent reviews to provide extensive information about the quality control analysis of ADCs (references 23 and 24, page 3 line 92).

Point 5. Masses of species with the corresponding errors (value ± SD) are presented in not good way; two significant digits of SD should be taken into consideration to present the results. Please, correct where required.

Masses indicated along the manuscript were obtained as an average of masses calculated for each charge state. As we are working in native conditions (not denaturing ones) on a Q-TOF instrument, it would not be significant to express mass accuracies with decimals. For instance, a SD of 0.1 Da in a mass measurement of an intact mAb or ADC represents less than 1 ppm of mass error. Typically, the associated error of an intact mass measurement of mAbs under native conditions is around 50-100 ppm, corresponding to 7-15 Da. For this reason, we do not consider decimals in the case of SD of mAb mass measurements at the intact level.

Round 2

Reviewer 3 Report

I do not have future comments